# Thermodynamic Restrictions in Linear Viscoelasticity

**DOI:** 10.3390/ma15082706

**Published:** 2022-04-07

**Authors:** Angelo Morro

**Affiliations:** DIBRIS, Università di Genova, 16145 Genova, Italy; angelo.morro@unige.it

**Keywords:** viscoelasticity, thermodynamic restrictions, strain–rate histories, free energy, 74D05, 74A20, 74A15, 80A17

## Abstract

The thermodynamic consistency of linear viscoelastic models is investigated. First, the classical Boltzmann law of stress–strain is considered. The kernel (Boltzmann function) is shown to be consistent only if the half-range sine transform is negative definite. The existence of free-energy functionals is shown to place further restrictions. Next, the Boltzmann function is examined in the unbounded power law form. The consistency is found to hold if the stress functional involves the strain history, not the strain–rate history. The stress is next taken to be given by a fractional order derivative of the strain. In addition to the constitutive equations involving strain–rate histories, finding a free-energy functional, consistent with the second law, seems to be an open problem.

## 1. Introduction

The modelling of materials with memory through functionals on an appropriate set of histories shows interesting questions about the correct assumptions on the constitutive equations. This is the case even for the linear theory of viscoelasticity within the realm of rational thermodynamics.

The classical linear theory traces back to Boltzmann [1,2] and assumes that the stress is determined (linearly) by the present value of the strain and the strain history. The consistency of this model with the second law of thermodynamics is well established. Here, we re-examine the thermodynamic restrictions and find that the kernel (Boltzmann function) is required to have a negative half-range sine transform. Well-known forms of the free-energy functional are shown to be consistent with thermodynamics, only if additional conditions on the kernel hold.

There are models in linear viscoelasticity where the kernel is unbounded [3]. This is particularly the case of kernels in a power law form [4,5,6]. We examine the thermodynamic consistency and show that the power law form is allowed if the exponent of the kernel is within (−1,0) and is applied to the strain history.

The modelling of viscoelasticity, as developed by Pipkin [7], involves the strain–rate history rather than the strain history. The strain–rate dependence may be justified as more appropriate to represent the continuity of the stress functional in that small changes in the strain–rate history produce small changes in the stress. In addition, we might think that the viscous character of viscoelasticity is more properly described by the strain–rate history. Mathematical difficulties arise if the strain–rate history is involved along with an unbounded kernel.

The power law form of the kernel is also characteristic of viscoelastic models with derivatives of fractional order. Fractional calculus is a well-established scheme in engineering science, particularly in materials modelling. Despite the extensive literature on the applications of derivatives of fractional order (see, e.g., [8,9,10] and Refs therein), it seems that no definite thermodynamic analysis has been developed so far. Here, the thermodynamic consistency is investigated by requiring the compatibility—with the second law—of a linear dependence of the stress on a fractional derivative of the strain and the existence of a corresponding free-energy functional.

*Notation*. We consider a solid occupying a time-dependent region $\Omega \subset E^{3}$. Throughout, ρ is the mass density, v is the velocity, T is the symmetric stress tensor, ε is the internal energy, and q is the heat flux. The symbol *∇* denotes the gradient operator in Ω, and $\partial_{t}$ is the partial time derivative at a point x∈Ω, while a superposed dot stands for the total time derivative, $\dot{f}=\partial_{t}f+v\cdot \nabla f$. Cartesian coordinates are used, L is the velocity gradient, $L_{ij}=\partial_{x_{j}}v_{i}$, and D=symL is the Eulerian rate of deformation. We let R be a reference configuration; the motion χ is a function that maps each point vector X∈R into a point x∈Ω. The deformation gradient F is defined by $F_{iK}=\partial_{X_{K}}\chi_{i}$ and J=detF>0, while $\nabla {\;}_{R}\;$ is the gradient in R. The Green–St. Venant strain is $E=\frac{1}{2}\left(F^{T}F-1\right)$, where 1 is the unit second-order tensor. If A is a second-order or fourth-order tensor, then the inequality A≥0 (or A≤0) means that A is positive (or negative) semi-definite.

## 2. Linear Viscoelastic Models

Throughout, we follow a Lagrangian formulation. Let $T_{RR}$ be the second Piola stress related to the Cauchy stress T, by
(1)$$T_{RR}=JF^{-1}TF^{-T}.$$

We then take the (linear) Boltzmann law [1] in the form
(2)$$T_{RR}\left(t\right)=G_{0}E\left(t\right)+\int_{0}^{\infty }G^{\prime }\left(s\right)E\left(t-s\right)ds,$$
where the common dependence on the point X∈R is understood and not written. Here, $G_{0}$ and $G^{\prime }\left(s\right)$ have values in the space Lin(Sym, Sym) of fourth-order tensors from Sym to Sym. We let
(3)$$G\left(s\right)=G_{0}+\int_{0}^{s}G^{\prime }\left(u\right)du,\;G_{\infty }=G_{0}+\int_{0}^{\infty }G^{\prime }\left(u\right)du.$$

We assume $G^{\prime }$ and $G^{\prime \prime }$ are continuous on [0,∞). Moreover we let $∥G^{\prime }∥={\left(G^{\prime }\cdot G^{\prime }\right)}^{1/2}$ and assume
(4)$$\int_{0}^{\infty }∥G^{\prime }∥\left(s\right)ds<\infty .$$

Since we use the Lagrangian formulation the dependence on the position is through the reference position X. Hence, both a superposed dot and $\partial_{t}$ denote the total time derivative, namely the derivative with X fixed.

## 3. Second Law and Free-Energy Functionals

Let θ be the absolute temperature, η be the entropy density per unit mass, $q_{R}$ be the heat flux in R, *r* be the energy supply, and $\rho_{R}$ be the mass density in R. Following the approach of rational thermodynamics, we take the balance of energy in the following form:(5)$$\rho_{R}\dot{\varepsilon }=T_{RR}\cdot \dot{E}-\nabla {\;}_{R}\;\cdot q_{R}+\rho_{R}r$$
and the balance of entropy in the following form:(6)$$\rho_{R}\dot{\eta }\geq \frac{\rho_{R}r}{\theta }-\nabla {\;}_{R}\;\cdot \frac{q_{R}}{\theta }.$$

We assume, as with the statement of the second law of thermodynamics, that inequality (Equation 6) holds for any admissible process; that is, for any set of admissible constitutive equations satisfying the balance equations. By replacing $\rho_{R}r-\nabla {\;}_{R}\;\cdot q_{R}$ from the balance of energy and considering the Helmholtz free energy ψ=ε−θη, we obtain the second-law inequality in the following form:(7)$$-\rho_{R}\left(\dot{\psi }+\eta \dot{\theta }\right)+T_{RR}\cdot \dot{E}-\frac{1}{\theta }q_{R}\cdot \nabla {\;}_{R}\;\theta \geq 0.$$

It is worth remarking that here we follow the scheme of continuum mechanics merely because this is the customary framework of viscoelasticity. Thermoviscous properties might be framed within a more general scheme by enlarging the notion of state with appropriate internal variables, as is the common case in extended irreversible thermodynamics [11,12]. Moreover, nonequilibrium properties might be described by means of rate-type equations, as is the case, e.g., in [13]. We also observe that inequality (Equation 7) shows that occurrence of flux–force pairs, such as $\dot{E},T_{RR}$ and $\nabla {\;}_{R}\;\theta ,q_{R}$. This is not generally the case, as is shown by balances derived within microscopic statistical approaches [12,14].

As a further simplifying assumption, we let θ be uniform, $\nabla {\;}_{R}\;\theta =0$, so that some properties of the free energy are conserved while accounting for equilibrium thermal processes. Hence, inequality (Equation 7) simplifies to
(8)$$-\rho_{R}\left(\dot{\psi }+\eta \dot{\theta }\right)+T_{RR}\cdot \dot{E}\geq 0.$$

Furthermore, with reference to the Boltzmann law, let $\psi ,\eta ,T_{RR}$, at time *t*, be dependent on the present values θ(t),E(t), and the history $E^{t}$,
(9)$$\psi \left(t\right)=\hat{\psi }\left(\theta \left(t\right),E\left(t\right),E^{t}\right).$$

Let $t_{0}$ be any time and let θ,E be constant at all times subsequent to $t_{0}$. Formally, given $t,t_{0}$, with $t>t_{0}$, consider the static continuation of $E^{t_{0}}$,
$$E^{t}\left(s\right)=\{\begin{array}{cc} E(t_{0}), & s\in [0,t-t_{0}],\\ E^{t_{0}}\left(s\right), & s\in (t-t_{0},\infty ), \end{array}$$
while θ is constant on $\left[t_{0},t\right]$. Hence, $\dot{\theta }=0$ and $\dot{E}=0$ on $\left[t_{0},t\right]$. Consequently, integration of (Equation 8) on $\left[t_{0},t\right]$ yields
(10)$$\hat{\psi }\left(\theta \left(t_{0}\right),E\left(t\right),E^{t}\right)-\hat{\psi }\left(\theta \left(t_{0}\right),E\left(t_{0}\right),E^{t_{0}}\right)\leq 0.$$

Assume that the functional (Equation 9) satisfies
(11)$$t\to \infty \;⟹\;\hat{\psi }\left(E^{t}\right)\to \hat{\psi }\left(E^{†}\right),$$
where $E^{†}$ is the constant history $E^{t}\left(s\right)=E\left(t_{0}\right),s\in \left[0,\infty \right)$, and then
(12)$$\hat{\psi }\left(E^{†}\right)\leq \hat{\psi }\left(E^{t_{0}}\right);$$
for brevity, we omit writing the dependence on the present values θ(t),E(t).

The result,
(13)$$\hat{\psi }\left(E^{t}\right)\geq \hat{\psi }\left(E^{†}\right)$$
means that among the free energies of the histories $E^{t}$ with a given present value, that associated with the constant history has the minimum value. This conclusion holds, irrespective of the linearity of the model.

In light of the dependence on $\theta ,E,E^{t}$, inequality (Equation 7) becomes
$$-\rho_{R}\left(\partial_{\theta }\hat{\psi }+\eta \right)\dot{\theta }+\left(T_{RR}-\rho_{R}\partial_{E}\hat{\psi }\right)\cdot \dot{E}-\rho_{R}d\hat{\psi }\left(E^{t}|{\dot{E}}^{t}\right)\geq 0,$$
where $d\hat{\psi }\left(E^{t}|{\dot{E}}^{t}\right)$ denotes the Fréchet derivative at $E^{t}$ in the direction of ${\dot{E}}^{t}$. The linearity and arbitrariness of $\dot{\theta }$ and $\dot{E}$ imply
(14)$$\eta =-\partial_{\theta }\hat{\psi },\;T_{RR}=\rho_{R}\partial_{E}\hat{\psi },\;d\hat{\psi }\left(E^{t}|{\dot{E}}^{t}\right)\leq 0.$$

These relations hold for any functional $\hat{\psi }\left(\theta ,E,E^{t}\right)$; to save writing we let θ and E stand for θ(t) and E(t).

If $\hat{\psi }$ is independent of $E^{t}$, then $T_{RR}=\rho_{R}\partial_{E}\hat{\psi }$ is the standard relation characterizing hyperelasticity. In such a case, it is often assumed that the reference configuration is natural ([15], §48.2.3) in that $\partial_{E}\hat{\psi }=0$ and
(15)$$\partial_{E}T_{RR}=\partial_{E}^{2}\hat{\psi }\geq 0$$
at E=0. The requirement (Equation 15) is viewed as the convexity condition and allows for the invertibility of $T_{RR}\left(E\right)$. A property of $\partial_{E}^{2}\hat{\psi }$ is related to wave propagation. If we look for jump discontinuities $a=[\;\;\;[\partial_{n}u]\;\;\;]$, in the direction n, by the equation of motion $\rho_{R}\ddot{u}=\nabla_{R}\cdot \left(FT_{RR}\right)$, at F=1, then we find that
(16)$$a\cdot A\left(n\right)a=\rho_{R}U^{2}{\left|a\right|}^{2}$$
where *U* is the speed and A(n) is the acoustic tensor, $A\left(n\right)=n\partial_{E}^{2}\hat{\psi }n$. To guarantee wave propagation the tensor $\partial_{E}^{2}\hat{\psi }$ is assumed to be strongly elliptic ([16], ch. 11),
(17)$$\left(a⊗n\right)\cdot \partial_{E}^{2}\hat{\psi }\left(a⊗n\right)>0$$
for all a,n. In the linear case, $\partial_{E}T_{RR}=G_{0}\in$ Lin(Sym, Sym) and $G_{0}>0$ satisfies the propagation condition (Equation 17).

We now go back to linear viscoelasticity and specify $T_{RR}$ in the Boltzmann form (Equation 2). By (Equation 14) we find
$$\rho_{R}\partial_{E}\hat{\psi }=G_{0}E+\int_{0}^{\infty }G^{\prime }\left(s\right)E^{t}\left(s\right)ds,$$
whence, if $G_{0}=G_{0}^{T}$,
(18)$$\rho_{R}\hat{\psi }\left(\theta ,E,E^{t}\right)=\rho_{R}\psi_{0}\left(\theta \right)+\frac{1}{2}E\cdot G_{0}E+E\cdot \int_{0}^{\infty }G^{\prime }\left(s\right)E^{t}\left(s\right)ds+\rho_{R}\tilde{\psi }\left(\theta ,E^{t}\right),$$
where $G_{0}$ and $G^{\prime }$ possibly are parameterized by the temperature θ. The functional $\tilde{\psi }$ is subject to two conditions. First, by (Equation 14)${}_{3}$, we have
(19)$$0\geq \rho_{R}d\hat{\psi }\left(E^{t}|{\dot{E}}^{t}\right)=E\cdot \int_{0}^{\infty }G^{\prime }\left(s\right){\dot{E}}^{t}\left(s\right)ds+\rho_{R}d\tilde{\psi }\left(E^{t}|{\dot{E}}^{t}\right).$$

Secondly, by (Equation 13) it follows that
(20)$$E\left(t\right)\cdot \int_{0}^{\infty }G^{\prime }\left(s\right)\left[E^{t}\left(s\right)-E\left(t\right)\right]ds+\tilde{\psi }\left(\theta ,E^{t}\right)-\tilde{\psi }\left(\theta ,E^{†}\right)\geq 0.$$

Inequalities (Equation 19) and (Equation 20) are necessary conditions on the free energy for the validity of the Boltzmann law (Equation 2).

Consider the functional
(21)$$\rho_{R}\psi_{G}=\rho_{R}\psi_{0}\left(\theta \right)+\frac{1}{2}E\cdot G_{\infty }E-\frac{1}{2}\int_{0}^{\infty }\left[E\left(t\right)-E^{t}\left(s\right)\right]\cdot G^{\prime }\left(s\right)\left[E\left(t\right)-E^{t}\left(s\right)\right]ds$$
and investigate the consistency with the requirements (Equation 19) and (Equation 20). We first observe that $\rho_{R}\psi_{G}$ is in the form (Equation 18). Splitting the dependence on E(t) and that on $E^{t}$ we have
(22)$$\begin{array}{c} \rho_{R}\psi_{G}=\rho_{R}\psi_{0}\left(\theta \right)+\frac{1}{2}E\cdot G_{\infty }E-\frac{1}{2}E\cdot \left(\int_{0}^{\infty }G^{\prime }\left(s\right)ds\right)E+E\cdot \int_{0}^{\infty }G^{\prime }\left(s\right)E^{t}\left(s\right)ds\\ -\frac{1}{2}\int_{0}^{\infty }E^{t}\left(s\right)\cdot G^{\prime }\left(s\right)E^{t}\left(s\right)\;ds. \end{array}$$

Since $\int_{0}^{\infty }G^{\prime }\left(s\right)ds=G_{\infty }-G_{0}$, it follows that $\psi_{G}$ has the form (Equation 18) with
(23)$$\rho_{R}\tilde{\psi }\left(\theta ,E^{t}\right)=-\frac{1}{2}\int_{0}^{\infty }E^{t}\left(s\right)\cdot G^{\prime }\left(s\right)E^{t}\left(s\right)\;ds.$$

At constant histories, namely when $E^{t}\left(s\right)=E\left(t\right)\;\forall s\geq 0$, we have
(24)$$\rho_{R}\psi_{G}=\rho_{R}\psi_{0}\left(\theta \right)+\frac{1}{2}E\cdot G_{\infty }E.$$

Hence, it follows that $\psi_{G}$ has a minimum value at constant histories if—and only if— the following holds
(25)$$\int_{0}^{\infty }\left[E\left(t\right)-E^{t}\left(s\right)\right]\cdot G^{\prime }\left(s\right)\left[E\left(t\right)-E^{t}\left(s\right)\right]ds\leq 0$$
where
(26)$$G^{\prime }\left(s\right)\leq 0\;\forall s\in \left[0,\infty \right).$$

Now, assuming $G^{\prime }=G^{\prime T}$ we have
(27)$$\rho_{R}d\psi_{G}\left(E^{t}|{\dot{E}}^{t}\right)=\int_{0}^{\infty }{\dot{E}}^{t}\left(s\right)\cdot G^{\prime }\left(s\right)\left[E\left(t\right)-E\left(t-s\right)\right]ds.$$

Observe that ${\dot{E}}^{t}\left(s\right)=-\partial_{s}E\left(t-s\right)=\partial_{s}\left[E\left(t\right)-E\left(t-s\right)\right]$. Moreover, letting
E(t,s):=E(t)−E(t−s)
we have
$$\partial_{s}\left\{E\left(t,s\right)\cdot G^{\prime }\left(s\right)E\left(t,s\right)\right\}=2\partial_{s}\left\{E\left(t,s\right)\right\}\cdot G^{\prime }\left(s\right)E\left(t,s\right)+E\left(t,s\right)\cdot G^{\prime \prime }\left(s\right)E\left(t,s\right).$$

Hence, an integration by parts yields the following: $$\rho_{R}d\psi_{G}=\frac{1}{2}{\left[E\left(t,s\right)\cdot G^{\prime }\left(s\right)E\left(t,s\right)\right]}_{0}^{\infty }-\frac{1}{2}\int_{0}^{\infty }E\left(t,s\right)\cdot G^{\prime \prime }\left(s\right)E\left(t,s\right)ds.$$

Since $G^{\prime }\left(\infty \right)=0$ and E(t,0)=0, the boundary terms (at s=0,∞) vanish. Hence, inequality (Equation 14)${}_{3}$ holds for any history $E^{t}$ if—and only if—the following holds:(28)$$G^{\prime \prime }\left(s\right)\geq 0\;\forall s\in \left[0,\infty \right).$$

Thus, the conditions (Equation 26) and (Equation 28) are necessary and sufficient for the consistency of the free-energy functional $\psi_{G}$.

The functional $\psi_{G}$ for the free energy traces back to Volterra [17,18]. The thermodynamic consistency of $\psi_{G}$ has been investigated by Graffi [19,20].

The results (Equation 26) and (Equation 28) have been obtained in the literature through various approaches; see, e.g., [21], where scalar-valued relaxation functions are considered. It is worth emphasizing that these restrictions follow up on the selection of a (nonunique) free-energy functional, as is the case also for the next example.

As with the previous scheme, let $G_{0}-G_{\infty }>0$. A further free-energy functional satisfying (Equation 18) is
(29)$$\rho_{R}\psi_{D}\left(\theta ,E,E^{t}\right)=\rho_{R}\psi_{0}\left(\theta \right)+\frac{1}{2}E\cdot G_{\infty }E+\Psi \left(E,E^{t}\right),$$
where
(30)$$\Psi \left(E,E^{t}\right)=\frac{1}{2}\int_{0}^{\infty }G^{\prime }\left(s\right)\left[E\left(t\right)-E^{t}\left(s\right)\right]ds\cdot {\left(G_{0}-G_{\infty }\right)}^{-1}\int_{0}^{\infty }G^{\prime }\left(s\right)\left[E\left(t\right)-E^{t}\left(s\right)\right]ds.$$

To verify the thermodynamic consistency of the functional (Equation 29), we first observe that the minimum property of $\psi_{D}$ at constant histories is apparent. Next, we note that
$$\begin{array}{c} \rho_{R}\psi_{D}\left(\theta ,E,E^{t}\right)=\rho_{R}\psi_{0}\left(\theta \right)+\frac{1}{2}E\cdot G_{\infty }E-\frac{1}{2}E\cdot \left(G_{\infty }-G_{0}\right)+E\cdot \int_{0}^{\infty }G^{\prime }\left(s\right)E^{t}\left(s\right)ds\\ +\frac{1}{2}\int_{0}^{\infty }G^{\prime }\left(s\right)E^{t}\left(s\right)ds\cdot {\left(G_{0}-G_{\infty }\right)}^{-1}\int_{0}^{\infty }G^{\prime }\left(s\right)E^{t}\left(s\right)ds. \end{array}$$

Hence, the form (Equation 18) holds with
$$\rho_{R}\tilde{\psi }=\frac{1}{2}\int_{0}^{\infty }G^{\prime }\left(s\right)E^{t}\left(s\right)ds\cdot {\left(G_{0}-G_{\infty }\right)}^{-1}\int_{0}^{\infty }G^{\prime }\left(s\right)E^{t}\left(s\right)ds.$$

Furthermore, we verify the requirement $d\psi_{D}\left(E^{t}|{\dot{E}}^{t}\right)\leq 0$. Since
$$\rho_{R}d\psi_{D}\left(E^{t}|{\dot{E}}^{t}\right)=\int_{0}^{\infty }G^{\prime }\left(s\right){\dot{E}}^{t}\left(s\right)ds\cdot {\left(G_{0}-G_{\infty }\right)}^{-1}\int_{0}^{\infty }G^{\prime }\left(s\right)\left[E\left(t\right)-E^{t}\left(s\right)\right]ds$$
an integration by parts yields
$$\begin{array}{c} \rho_{R}d\psi_{D}\left(E^{t}|{\dot{E}}^{t}\right)=\left\{{\left[G^{\prime }\left(s\right)\left[E\left(t\right)-E^{t}\left(s\right)\right]\right]}_{0}^{\infty }-\int_{0}^{\infty }G^{\prime \prime }\left(s\right)\left[E\left(t\right)-E^{t}\left(s\right)\right]ds\right\}\\ \cdot {\left(G_{0}-G_{\infty }\right)}^{-1}\int_{0}^{\infty }G^{\prime }\left(s\right)\left[E\left(t\right)-E^{t}\left(s\right)\right]ds. \end{array}$$

The vanishing of the boundary terms implies that
$$\rho_{R}d\psi_{D}\left(E^{t}|{\dot{E}}^{t}\right)=-\int_{0}^{\infty }G^{\prime \prime }\left(s\right)\left[E\left(t\right)-E^{t}\left(s\right)\right]ds\cdot {\left(G_{0}-G_{\infty }\right)}^{-1}\int_{0}^{\infty }G^{\prime }\left(s\right)\left[E\left(t\right)-E^{t}\left(s\right)\right]ds.$$

Consequently, $d\psi_{D}\left(E^{t}|{\dot{E}}^{t}\right)\leq 0$ for any history $E^{t}$ if—and only if—the following holds: (31)$$G^{\prime \prime }\left(s\right){\left(G_{0}-G_{\infty }\right)}^{-1}G^{\prime }\left(s\right)\leq 0\;\forall s\in \left[0,\infty \right).$$

**Remark** **1.**
*It is worth emphasizing that the restrictions (Equation 14) hold for possibly nonlinear models. Inequalities for $G^{\prime },G^{\prime \prime }$ are related to linear models.*


## 4. Restrictions Induced by Periodic Histories

We now examine the restrictions on the Boltzmann function $G^{\prime }$ induced by a particular set of functions of θ and E. Consider functions θ(t) and E(t), such that θ(t+d) and E(t+d) for any time *t*. Consequently,
$$E^{t}\left(s\right)=E^{t+d}\left(s\right)\;\forall s\in \left[0,\infty \right)$$
and hence the history $E^{t}$ is periodic, with period *d*. This in turn implies that
$$\psi \left(t+d\right)=\hat{\psi }\left(\theta \left(t+d\right),E\left(t+d\right),E^{t+d}\right)=\hat{\psi }\left(\theta \left(t\right),E\left(t\right),E^{t}\right)=\psi \left(t\right),$$
where $\hat{\psi }$ is also periodic. Moreover, let $\eta =\hat{\eta }\left(\theta \right)$ and *N* be the integral of $\hat{\eta }\left(\theta \right)$, so that
$$\frac{dN(\theta (t))}{dt}=\eta \dot{\theta }.$$

Now, N(θ(t)) is periodic too, with period *d*. Hence, the integration of (Equation 8) over t∈[0,d] results in
(32)$$\int_{0}^{d}T_{RR}\cdot \dot{E}\;dt\geq 0;$$
and the same conclusion follows for any function $\eta (\theta ,E,E^{t})$, if attention is restricted to isothermal processes, then

$\dot{\theta }\equiv 0$. In view of (Equation 32) and (Equation 2), we have
(33)$$\int_{0}^{\infty }\dot{E}\left(t\right)\cdot G_{0}E\left(t\right)dt+\int_{0}^{d}\left\{\dot{E}\left(t\right)\cdot \int_{0}^{\infty }G^{\prime }\left(s\right)E^{t}\left(s\right)ds\right\}dt\geq 0,$$
for any periodic functions E with period *d*. Here, we do not assume the symmetry of $G_{0}$ and $G^{\prime }$.

To exploit the inequality (Equation 33), we consider harmonic strain tensor functions
$$E\left(t\right)=E_{1}\cos\omega t+E_{2}\sin\omega t,$$

$E_{1},E_{2}$ being arbitrary symmetric tensors; Hence, d=2π/|ω|. Substituting in (Equation 33), and integrating, we obtain
(34)$$\omega \left\{E_{2}\cdot \left[G_{c}^{\prime }-G_{c}^{\prime T}\right]E_{1}-E_{1}\cdot G_{s}^{\prime }E_{1}-E_{2}\cdot G_{s}^{\prime }E_{2}+E_{2}\cdot \left[G_{0}-G_{0}^{T}\right]E_{1}\right\}\geq 0,$$
where $G_{c}^{\prime }$ and $G_{s}^{\prime }$ are the ω-dependent cosine and sine transforms of $G^{\prime }$; they are defined on R by
$$G_{c}^{\prime }\left(\omega \right)=\int_{0}^{\infty }G^{\prime }\left(u\right)\cos\omega u\;du,\;G_{s}^{\prime }\left(\omega \right)=\int_{0}^{\infty }G^{\prime }\left(u\right)\sin\omega u\;du,\;\omega \in R.$$

Let ω→∞. By Riemann’s lemma, it follows that
$$G_{c}^{\prime }\left(\omega \right)\to 0,\;G_{s}^{\prime }\left(\omega \right)\to 0.$$

Inequality (Equation 34) then reduces to
(35)$$E_{2}\cdot \left[G_{0}-G_{0}^{T}\right]E_{1}\geq 0.$$

First, inequality (Equation 35) holds, e.g., for any $E_{2}$ with $E_{1}$ and $-E_{1}$, only if $E_{2}\cdot \left[G_{0}-G_{0}^{T}\right]E_{1}=0$ for any pair $E_{2},E_{1}$. Next, the arbitrariness of $E_{1},E_{2}$ implies that $G_{0}-G_{0}^{T}=0$, namely
(36)$$G_{0}=G_{0}^{T}.$$

We now let ω→0. As we show in a while,
(37)$$\lim_{\omega \to 0}G_{s}^{\prime }\left(\omega \right)=0,\;\lim_{\omega \to 0}G_{c}^{\prime }\left(\omega \right)=\int_{0}^{\infty }G^{\prime }\left(u\right)du=G_{\infty }-G_{0}.$$

Hence, taking the limit of (Equation 34) as ω→0 we find
(38)$$G_{\infty }=G_{\infty }^{T}.$$

Finally, let $E_{1}=E_{2}=E$. Inequality (Equation 34) reduces to
$$\omega \;E\cdot G_{s}^{\prime }\left(\omega \right)E\leq 0$$
or
(39)$$\omega \;G_{s}^{\prime }\left(\omega \right)\leq 0.$$

The requirements (Equation 36)–(Equation 39) are necessary for the consistency of (Equation 2) with the second law of thermodynamics. By having recourse to Fourier series, we can prove that they are also sufficient [2]. We now derive some consequences of (Equation 39).

By the inversion formula, we have
(40)$$G^{\prime }\left(u\right)=\frac{2}{\pi }\int_{0}^{\infty }\sin\omega u\;G_{s}^{\prime }\left(\omega \right)d\omega .$$

Integration of (Equation 40) with respect to *u* yields
(41)$$G\left(u\right)-G_{0}=\frac{2}{\pi }\int_{0}^{\infty }\frac{1-\cos\omega u}{\omega }G_{s}^{\prime }\left(\omega \right)d\omega .$$

Inequality (Equation 40) then implies
(42)$$G_{0}-G\left(u\right)\geq 0\;\forall u\in \left[0,\infty \right).$$

Inequality (Equation 42) means that G(u) has a maximum at u=0 or that the instantaneous elastic modulus $G_{0}$ is the maximum value of G(u). However, this need not imply that G is monotone, decreasing as we might expect. It follows from (Equation 40) that, if $G_{s}^{\prime }\left(u\right)=\hat{G}g\left(u\right)$ and *g* is monotone decreasing, then $G^{\prime }$ is negative, and G is monotone decreasing.

If $G^{\prime \prime }\in L_{1}\left(R^{+}\right)$, then an integration by parts yields
$$G_{s}^{\prime }\left(\omega \right)=\int_{0}^{\infty }G^{\prime }\left(u\right)\sin\omega u\;du=-{\left[G^{\prime }\left(u\right)\frac{\cos\omega u}{\omega }\right]}_{0}^{\infty }+\frac{1}{\omega }\int_{0}^{\infty }G^{\prime \prime }\left(u\right)\cos\omega u\;du.$$

Hence, we have
$$\omega G_{s}^{\prime }\left(\omega \right)=G^{\prime }\left(0\right)+G_{c}^{\prime \prime }\left(\omega \right),$$
where $G^{\prime }\left(0\right)$ stands for $G^{\prime }\left(0_{+}\right)$. Consequently,
(43)$$\lim_{\omega \to \infty }\omega G_{s}^{\prime }\left(\omega \right)=G^{\prime }\left(0\right).$$

Equation (Equation 43) in turn implies that
$$G^{\prime }\left(0\right)\leq 0.$$

By the same token, we have
$$G_{c}^{\prime }\left(\omega \right)=-\frac{1}{\omega }G_{s}^{\prime \prime }\left(\omega \right)$$
and then
(44)$$\lim_{\omega \to \infty }\omega G_{c}^{\prime }\left(\omega \right)=0,\;G_{c}^{\prime }\left(\omega \right)=o\left(1/\omega \right).$$

### 4.1. Proof of (37)

By (Equation 3), it follows that for any arbitrarily small ϵ>0 there is $u_{0}$, such that
(45)$$\int_{u_{0}}^{\infty }∥G^{\prime }\left(u\right)∥du<\epsilon .$$

Inequality (Equation 45) implies
$$∥\int_{u_{0}}^{\infty }G^{\prime }\left(u\right)\sin\omega u\;du∥\leq \int_{u_{0}}^{\infty }∥G^{\prime }\left(u\right)∥\;|\sin\omega u|du<\epsilon .$$

Now, since |sinωu|≤|ωu|, we have
$$∥\int_{0}^{u_{0}}G^{\prime }\left(u\right)\sin\omega u\;du∥\leq \int_{0}^{u_{0}}∥G^{\prime }\left(u\right)∥\;|\sin\omega u|\;du\leq \left|\omega \right|u_{0}\int_{0}^{u_{0}}∥G^{\prime }\left(u\right)∥\;du\to 0$$
as ω→0. Consequently, $\lim_{\omega \to 0}G_{s}^{\prime }\left(\omega \right)=0$.

Likewise, observe that
$$\begin{array}{c} G_{c}^{\prime }\left(\omega \right)=\int_{0}^{\infty }G^{\prime }\left(u\right)\cos\omega u\;du=\int_{0}^{\infty }G^{\prime }\left(u\right)\left(\cos\omega u-1\right)du+\int_{0}^{\infty }G^{\prime }\left(u\right)\;du\\ =\int_{0}^{\infty }G^{\prime }\left(u\right)\left(\cos\omega u-1\right)du+G_{\infty }-G_{0}. \end{array}$$

For any ϵ>0, there is $u_{0}$, such that
$$∥\int_{u_{0}}^{\infty }G^{\prime }\left(u\right)\left(\cos\omega u-1\right)du∥\leq 2\int_{u_{0}}^{\infty }∥G^{\prime }\left(u\right)∥\;du<\epsilon .$$

Since |cosωu−1|≤(4/π)|ω|u then
$$∥\int_{0}^{u_{0}}G^{\prime }\left(u\right)\left(\cos\omega u-1\right)du∥\leq \left(4/\pi \right)|\omega |u_{0}\int_{0}^{u_{0}}∥G^{\prime }\left(u\right)∥du\to 0$$
as ω→0.

### 4.2. Remarks about the Half-Range Sine Transform

It seems reasonable to assume that G(u) enjoys the same tensor properties for any u∈[0,∞). Hence, we let
$$G\left(u\right)=G_{0}g\left(u\right),\;G_{0}=G_{0}^{T}>0.$$
and (Equation 2) becomes
$$T_{RR}\left(t\right)=G_{0}\{E\left(t\right)+\int_{0}^{\infty }g^{\prime }\left(u\right)E\left(t-u\right)du\}.$$

The restriction to periodic histories implies
(46)$$g_{s}^{\prime }\left(\omega \right)=\int_{0}^{\infty }\sin\omega u\;g^{\prime }\left(u\right)du\leq 0\;\forall \omega \geq 0,$$
and hence
(47)$$g^{\prime }\left(0\right)\leq 0,\;g_{s}^{\prime }\left(\omega \right)=O\left(\omega \right),\;g_{c}^{\prime }\left(\omega \right)=o\left(1/\omega \right).$$

A free energy satisfies the second law, if $g^{\prime }\left(u\right)\leq 0,g^{\prime \prime }\left(u\right)\geq 0\;\forall u>0$.

We now show that $g^{\prime }\left(u\right)\leq 0\;\forall u\geq 0⟹g_{s}^{\prime }\left(\omega \right)\leq 0\;\forall \omega \geq 0$ by means of the following:

**Lemma** **1.**
*Let $f:R^{+}\to R^{+}$. If f is continuous and monotone decreasing on $R^{+}$ and f(u)→0 as u→∞, then*

$$f_{s}\left(\omega \right)\geq 0\;\forall \omega \geq 0.$$



To prove (Lemma 1) we first observe that, for any ω>0, we can partition $R^{+}$ into subintervals [2jπ/ω,2(j+1)π/ω], j=0,1,2,… For any subinterval, we have
(48)$$\begin{array}{c} \int_{2j\pi /\omega }^{2(j+1)\pi /\omega }f\left(u\right)\sin\omega u\;du=\int_{2j\pi /\omega }^{2(j+1/2)\pi /\omega }f\left(u\right)\sin\omega u\;du+\int_{2(j+1/2)\pi /\omega }^{2(j+1)\pi /\omega }f\left(u\right)\sin\omega u\;du\\ \leq \left[f\left(\frac{2j\pi }{\omega }\right)-f\left(\frac{2(j+1)\pi }{\omega }\right)\right]\int_{0}^{\pi /\omega }\sin\omega u\;du=\frac{2}{\omega }\left[f\left(\frac{2j\pi }{/\omega }\right)-f\left(\frac{2(j+1)\pi }{\omega }\right)\right]\geq 0 \end{array}$$
and
(49)$$\begin{array}{c} \int_{2j\pi /\omega }^{2(j+1)\pi /\omega }f\left(u\right)\sin\omega u\;du\;\;\;\;\;\\ \geq f\left(\frac{2(j+1/2)\pi }{\omega }\right)[\int_{2j\pi /\omega }^{2(j+1/2)\pi /\omega }\sin\omega u\;du+\int_{2(j+1/2)\pi /\omega }^{2(j+1)\pi /\omega }\sin\omega u\;du]=0. \end{array}$$

Inequalities (Equation 48) and (Equation 49) imply
$$0\leq \sum_{j=0}^{n-1}\int_{2j\pi /\omega }^{2(j+1)\pi /\omega }f\left(u\right)\sin\omega u\;du\leq \frac{2}{\omega }\left[f\left(0\right)-f\left(2n\pi /\omega \right)\right].$$

Now, for any $\tilde{u}\in R^{+}$ let $n=\left\lfloor \tilde{u}/2\pi /\omega \right\rfloor$ and observe that
$$0\leq \int_{2n\pi /\omega }^{\tilde{u}}f\left(u\right)\sin\omega u\;du\leq \frac{2}{\omega }f\left(2n\pi /\omega \right).$$

Consequently, for any $\tilde{u}>0$, we have
(50)$$0\leq \int_{0}^{\tilde{u}}f\left(u\right)\sin\omega u\;du\leq \frac{2}{\omega }f\left(0\right).$$

For any ω>0, taking the limit of (Equation 50) as $\tilde{u}\to \infty$ we obtain
$$0\leq f_{s}\left(\omega \right)\leq \frac{2}{\omega }f\left(0\right),$$
while, by definition, $f_{s}\left(0\right)=0$. This concludes the proof.

Applying the Lemma to $f=-g^{\prime }\geq 0$, we obtain $-g_{s}^{\prime }\left(\omega \right)\geq 0\;\forall \omega \geq 0$, and hence (Equation 46).

## 5. Examples of Boltzmann Functions

### Prony Series

Perhaps the most widely used form of Boltzmann function is the so-called Prony series, namely a linear superposition of decreasing exponentials,
$$g^{\prime }\left(u\right)=\sum_{k=1}^{N}c_{k}\exp\left(-\alpha_{k}u\right),\;c_{k}<0,\;\alpha_{k}>0,$$
where $1/\alpha_{k}$ may be viewed as the *k*-th relaxation time. Hence,
$$g^{\prime }\left(0\right)=\sum_{k=1}^{N}c_{k}<0.$$

Apparently, $g^{\prime }$ is a bounded monotone decreasing function on [0,∞). Moreover, since
$$\int_{0}^{\infty }\exp\left(-\alpha_{k}u\right)\exp\left(i\omega u\right)du={\left[\frac{\exp\{-\alpha_{k}+i\omega )u\}}{-\alpha_{k}+i\omega }\right]}_{0}^{\infty }=\frac{\alpha_{k}+i\omega }{\alpha_{k}^{2}+\omega^{2}}$$
then
$$g_{c}^{\prime }\left(\omega \right)=\sum_{k=1}^{N}\frac{c_{k}\alpha_{k}}{\alpha_{k}^{2}+\omega^{2}},\;g_{s}^{\prime }\left(\omega \right)=\sum_{i=1}^{N}\frac{c_{k}\omega }{\alpha_{k}^{2}+\omega^{2}}.$$

Accordingly, $g_{c}^{\prime }\left(\omega \right)<0$ and
$$g_{s}^{\prime }\left(\omega \right)<0\;\forall \omega \in \left(0,\infty \right).$$

Moreover, $g_{c}^{\prime }$ and $g_{s}^{\prime }$ satisfy (Equation 47).

According to the literature (see [4,8] and the Refs therein), it is of interest to consider Boltzmann functions in the following power law form:$$g^{\prime }\left(u\right)=-u^{-\beta },\;\beta \in \left(0,1\right).$$

The integral
(51)$$\int_{0}^{\infty }u^{-\beta }E\left(t-u\right)du$$
is bounded if $E^{t}$ has compact support, i.e., $E^{t}\left(u\right)=0,u\geq a$. The integral (Equation 51) is also bounded if $E^{t}$ has a harmonic dependence,
(52)$$E\left(t-u\right)=E_{1}\cos\omega \left(t-u\right)+E_{2}\sin\omega \left(t-u\right).$$

In the case of (Equation 52),
$$\int_{0}^{\infty }u^{-\beta }E\left(t-u\right)du=I_{\beta }\left(E_{1}\sin\omega t-E_{2}\cos\omega t\right)+J_{\beta }\left(E_{1}\cos\omega t+E_{2}\sin\omega t\right),$$
where
$$I_{\beta }=\int_{0}^{\infty }u^{-\beta }\sin\omega u\;du,\;J_{\beta }=\int_{0}^{\infty }u^{-\beta }\cos\omega u\;du.$$

Both $I_{\beta }$ and $J_{\beta }$ are bounded as β∈(0,1). Instead, if E is constant, then (Equation 51) diverges.

## 6. Viscoelastic Models with Strain–Rate Histories

Based on the observation that the viscoelastic behaviour is a combination of elastic and viscous effects, Pipkin [7] suggested that the strain–rate history should be involved rather than the strain history [22]. Hence the constitutive equation might be written in the form
(53)$$T_{RR}\left(t\right)=G_{0}E\left(t\right)+\int_{0}^{\infty }M\left(u\right)\dot{E}\left(t-u\right)du.$$

If M(0) is bounded, then an integration by parts and the assumption M(∞)=0 yield
(54)$$T_{RR}\left(t\right)=\left[G_{0}+M\left(0\right)\right]E\left(t\right)+\int_{0}^{\infty }M^{\prime }\left(u\right)E\left(t-u\right)du.$$

Indeed, $G_{0}+M\left(0\right)$ would be the instantaneous modulus and $M^{\prime }$ the Boltzmann function. In that case, the thermodynamic requirement is
(55)$$M_{s}^{\prime }\left(\omega \right)\leq 0\;\forall \omega \in \left[0,\infty \right).$$

Yet, it seems that the effect of ${\dot{E}}^{t}$ on the stress $T_{RR}$ is significant if we cannot pass to (Equation 54) because M(u) is unbounded as $u\to 0_{+}$. Suppose that we cannot integrate by parts and observe that in connection with the time-harmonic dependence
$$E\left(t\right)=E_{1}\cos\omega t+E_{2}\sin\omega t$$
we can repeat the procedure of §4 and require that the functional (Equation 53) satisfy
$$\int_{0}^{2\pi /\omega }T_{RR}\left(t\right)\cdot \dot{E}\left(t\right)dt\geq 0.$$

This requirement results in
$$\begin{array}{c} 0\leq \int_{0}^{2\pi /\omega }dt\{\left(-\omega E_{1}\sin\omega t+\omega E_{2}\cos\omega t\right)\cdot G_{0}\left(E_{1}\cos\omega t+E_{2}\sin\omega t\right)\;\;\\ +\left(-\omega E_{1}\sin\omega t+\omega E_{2}\cos\omega t\right)\cdot \int_{0}^{\infty }duM\left(u\right)\left[\left(-\omega E_{1}\sin\omega \left(t-u\right)+\omega E_{2}\cos\omega \left(t-u\right)\right)\right]\}. \end{array}$$
where it follows that
$$E_{1}\cdot \left(G_{0}^{T}-G_{0}\right)E_{2}+\omega \left[E_{1}\cdot M_{c}\left(\omega \right)E_{1}+E_{2}\cdot \left[M_{s}\left(\omega \right)-M_{s}^{T}\left(\omega \right)\right]E_{1}+E_{2}\cdot M_{c}\left(\omega \right)E_{2}\right]\geq 0.$$

Letting ω→∞, we find again $G_{0}=G_{0}^{T}$. Now, we let $E_{1}=E_{2}$, and obtain
(56)$$\omega M_{c}\left(\omega \right)\geq 0\;\forall \omega \in R.$$

Inequalities (Equation 55) and (Equation 56) are consistent in that, integrating by parts with M(0) bounded, we have
$$\omega M_{c}\left(\omega \right)=\omega \int_{0}^{\infty }M\left(u\right)\cos\omega u\;du={\left[M\left(u\right)\sin\omega u\right]}_{0}^{\infty }-\int_{0}^{\infty }M^{\prime }\left(u\right)\sin\omega u\;du=-M_{s}^{\prime }\left(\omega \right).$$

If we let $M\left(u\right)=M_{0}u^{-\beta }$, β∈(0,1), then M(u)sinωu→0 as u→0 and
$$M^{\prime }\left(u\right)\sin\omega u=-\beta M_{0}u^{-\beta }\frac{\sin\omega u}{u}$$
is integrable. Hence, $\omega M_{c}\left(\omega \right)=-M_{s}^{\prime }\left(\omega \right)$ also holds if $M\left(u\right)≃u^{-\beta }$.

As we show in the next section, in connection with the general context of models of fractional order, consistency with thermodynamics requires also that there exists a free-energy functional ψ of the form
$$\psi \left(E,{\dot{E}}^{t}\right)=\frac{1}{2}E\cdot G_{0}E+E\cdot \int_{0}^{\infty }M\left(u\right){\dot{E}}^{t}\left(u\right)du+\Psi \left({\dot{E}}^{t}\right)$$
with $d\psi (E,{\dot{E}}^{t}|{\ddot{E}}^{t})\leq 0.$ The existence of such a functional is investigated.

## 7. Viscoelastic Models of Fractional Order

Still with attention to both a power law form of the kernel and dependence of the stress on the strain–rate we may consider the following constitutive equation: (57)$$T_{RR}\left(t\right)=M\int_{-\infty }^{t}{\left(t-u\right)}^{-\beta }\partial_{u}E\left(u\right)du,$$

M being a dimensional quantity and most likely β∈(0,1). Differently from (Equation 53), we neglect the dependence on the present value E(t). Since we have in mind the standard notation for models with derivatives of fractional order, in this section, $\partial_{u}$ and $\partial_{t}$ denote the (total) time derivative at constant reference position X.

While the power law of data may be the physical motivation for assuming constitutive equations of the form (Equation 57), we observe that, if β∈(0,1), then ${\left(t-u\right)}^{-\beta }$ is not integrable on (−∞,t]. We then might restrict attention to the set of strain histories with compact support,
F={E(u):E(u)=0asu∈(−∞,a],E(u)∈Symasu∈[a,t]},

*a* being a suitable reference time (called base point in the literature) or to histories, such that (Equation 57) converges. This is the case for time-harmonic histories.

This view leads naturally to the modelling via fractional derivatives. In light of the Caputo fractional derivative [23], for any E∈F, we let
$$D_{t}^{\beta }E\left(t\right)=\frac{1}{\Gamma (1-\beta )}\int_{-\infty }^{t}{\left(t-u\right)}^{-\beta }\partial_{u}E\left(u\right)du$$
where Γ is the Gamma function,
$$\Gamma \left(1-\beta \right)=\int_{0}^{\infty }\exp\left(-t\right)t^{-\beta }dt.$$

By a change of variable, we have
$$D_{t}^{\beta }E\left(t\right)=\frac{1}{\Gamma (1-\beta )}\int_{0}^{\infty }u^{-\beta }\partial_{t}E\left(t-u\right)du.$$

Since E∈F, E(t−u)=0 as u>t−a. Instead of restricting the set of histories, we may replace the integral on [0,∞) with the integral on [0,a] [24].

Still, for functions in F, we define the fractional derivative of any order. Let n∈N and α∈[n−1,n). Let $\left\lfloor \alpha \right\rfloor$ denote the floor function of α, here $\left\lfloor \alpha \right\rfloor =n-1$. We define the derivative $D_{t}^{\alpha }$ in the form
$$D_{t}^{\alpha }E\left(t\right)=\frac{1}{\Gamma (\left\lfloor \alpha \right\rfloor +1-\alpha )}\int_{0}^{\infty }u^{-\alpha +\left\lfloor \alpha \right\rfloor }\partial_{t}^{\left\lfloor \alpha \right\rfloor +1}E\left(t-u\right)du.$$

For any α∈N we have $\alpha =\left\lfloor \alpha \right\rfloor$ and $\Gamma \left(\left\lfloor \alpha \right\rfloor +1-\alpha \right)=\Gamma \left(1\right)=1$. Consequently,
$$D_{t}^{\left\lfloor \alpha \right\rfloor }E\left(t\right)=\int_{0}^{\infty }\partial_{t}^{\left\lfloor \alpha \right\rfloor +1}E\left(t-u\right)du={\left[\partial_{t}^{\left\lfloor \alpha \right\rfloor }E\left(t-u\right)\right]}_{0}^{\infty }=\partial_{t}^{\left\lfloor \alpha \right\rfloor }E\left(t\right).$$

We let α≥0. For any $\alpha \in R^{+}\setminus N$ the fractional derivative $D_{t}^{\left\lfloor \alpha \right\rfloor }E\left(t\right)$ is a linear functional of the history $\partial_{t}^{\left\lfloor \alpha \right\rfloor +1}E^{t}$. If, instead, α∈N, then the fractional derivative coincides with the corresponding time derivative.

A simple example of constitutive equation of fractional order might be considered in the following form: (58)$$T_{RR}\left(t\right)=MD_{t}^{\alpha }E\left(t\right),$$
where M∈ Lin(Sym, Sym). Hence, we have
$$T_{RR}\left(t\right)=\frac{1}{\Gamma (\left\lfloor \alpha \right\rfloor +1-\alpha )}M\int_{0}^{\infty }u^{\left\lfloor \alpha \right\rfloor -\alpha }\partial_{t}^{\left\lfloor \alpha \right\rfloor +1}E\left(t-u\right)du.$$

For formal convenience, let
$$\tilde{M}\left(u\right)=\frac{1}{\Gamma (\left\lfloor \alpha \right\rfloor +1-\alpha )}Mu^{\left\lfloor \alpha \right\rfloor -\alpha }.$$

Hence, the constitutive Equation (Equation 58) can be written in the form
(59)$$T_{RR}\left(t\right)=\int_{0}^{\infty }\tilde{M}\left(u\right)\partial_{t}^{\left\lfloor \alpha \right\rfloor +1}E\left(t-u\right)du,$$
thus ascribing to $\tilde{M}$ the meaning of kernel, parametrized by the order α and possibly the temperature θ. If α∈N, then (Equation 58) becomes
(60)$$T_{RR}\left(t\right)=M\partial_{t}^{\alpha }E\left(t\right).$$

Let α∈(0,1) and hence $\left\lfloor \alpha \right\rfloor =0$. Equation (Equation 59) simplifies to
(61)$$T_{RR}\left(t\right)=\int_{0}^{\infty }\tilde{M}\left(u\right)\partial_{t}E\left(t-u\right)du.$$

Observe $\partial_{t}E\left(t-u\right)=-\partial_{u}E\left(t-u\right)$ and then, integrating by parts, we find
(62)$$T_{RR}\left(t\right)=\int_{0}^{\infty }\tilde{M}\left(u\right)\partial_{t}E\left(t-u\right)du=\tilde{M}\left(0\right)E\left(t\right)+\int_{0}^{\infty }{\tilde{M}}^{\prime }\left(u\right)E\left(t-u\right)du.$$

Equation (Equation 62) would be the classical form of the Boltzmann law. However $\tilde{M}\left(0\right)$ diverges and hence a different approach is in order.

**Remark** **2.**
*We may wonder about the thermodynamic consistency of constitutive equations of the form (Equation 58) with the second law of thermodynamics. Friedrich [25] considered the stress–strain relation in the form of the generalized Maxwell model*

$$\sigma \left(t\right)+\lambda^{\alpha }D_{t}^{\alpha }\sigma =\lambda^{\beta }ED_{t}^{\beta }\varepsilon ,$$

*where σ is the stress, E the spring constant, and ε the strain. It emerged that thermodynamic compatibility holds, or the solution is thermodynamically reasonable [26], if 1≥β≥α>0. The consistency with thermodynamics is now investigated in detail.*


### Fractional Models and Thermodynamic Requirements

Consider models where the constitutive functionals $\psi ,\eta ,q_{R}$, and $T_{RR}$, at time *t*, depend on the set of variables $\theta \left(t\right),E\left(t\right),\partial_{t}E\left(t\right),\partial_{t}^{\left\lfloor \alpha \right\rfloor +1}E^{t},\nabla {\;}_{R}\;\theta \left(t\right)$. The Clausius–Duhem inequality (Equation 7) yields
(63)$$\begin{array}{c} -\rho_{R}\left(\partial_{\theta }\psi +\eta \right)\partial_{t}\theta +\left(T_{RR}-\rho_{R}\partial_{E}\psi \right)\cdot \partial_{t}E-\rho_{R}\partial_{\partial_{t}E}\psi \cdot \partial_{t}^{2}E\;\;\\ -\rho_{R}d\psi \left(\partial_{t}^{\left\lfloor \alpha \right\rfloor +1}E^{t}|\partial_{t}^{\left\lfloor \alpha \right\rfloor +2}E^{t}\right)-\rho_{R}\partial_{\nabla {\;}_{R}\;\theta }\psi \cdot \nabla {\;}_{R}\;\partial_{t}\theta -\frac{1}{\theta }q_{R}\cdot \nabla {\;}_{R}\;\theta \geq 0, \end{array}$$
where $d\psi (\partial_{t}^{\left\lfloor \alpha \right\rfloor +1}E^{t}|\partial_{t}^{\left\lfloor \alpha \right\rfloor +2}E^{t})$ is the Fréchet derivative of ψ with respect to $\partial_{t}^{\left\lfloor \alpha \right\rfloor +1}E^{t}$ along $\partial_{t}^{\left\lfloor \alpha \right\rfloor +2}E^{t}$. The linearity and arbitrariness of $\nabla {\;}_{R}\;\partial_{t}\theta$, $\partial_{t}^{2}E$, $\partial_{t}\theta$ imply that
(64)$$\partial_{\nabla {\;}_{R}\;\theta }\psi =0,\;\partial_{\partial_{t}E}\psi =0,\;\eta =-\partial_{\theta }\psi .$$

Since ψ is independent of $\nabla {\;}_{R}\;\theta$, the remaining inequality implies that
(65)$$\left(T_{RR}-\rho_{R}\partial_{E}\psi \right)\cdot \partial_{t}E-\rho_{R}d\psi \left(\partial_{t}^{\left\lfloor \alpha \right\rfloor +1}E^{t}|\partial_{t}^{\left\lfloor \alpha \right\rfloor +2}E^{t}\right)\geq 0.$$

The heat conduction inequality
$$q_{R}\cdot \nabla {\;}_{R}\;\theta \leq 0$$
is consistent, though non-necessary, if $q_{R}$ depends on $\partial_{t}E\left(t\right),\partial_{t}^{\left\lfloor \alpha \right\rfloor +1}E^{t}$.

For any given history $E_{*}^{t}$, we can select a history $E^{t}$, such that $\partial_{t}E\left(t\right)$ is arbitrary, while $∥\partial_{E}\psi \left(E_{*}^{t}\right)-\partial_{E}\psi \left(E_{*}^{t}\right)∥$ is as small as we please; this is obtained by letting ψ be continuously differentiable and $E_{*}^{t}\left(u\right)-E^{t}\left(u\right)=0$ as u≥a with *a* arbitrarily small. Hence, it follows
(66)$$\left(T_{RR}-\rho_{R}\partial_{E}\psi \right)\cdot \partial_{t}E\geq 0,\;d\psi \left(\partial_{t}^{\left\lfloor \alpha \right\rfloor +1}E^{t}|\partial_{t}^{\left\lfloor \alpha \right\rfloor +2}E^{t}\right)\leq 0,\;q_{R}\cdot \nabla {\;}_{R}\;\theta \leq 0.$$

If we assume
$$\partial_{E}\psi =M_{0}E$$
then we have
$$T_{RR}=M_{0}E+M_{1}\partial_{t}E,\;\psi =\Psi \left(E\right)+\tilde{\psi }\left(\theta ,\partial_{t}^{\left\lfloor \alpha \right\rfloor +1}E^{t}\right),\;\Psi \left(E\right)=\frac{1}{2}E\cdot M_{0}E,$$
where $M_{1}$ is positive definite. This makes the relation for $T_{RR}$ as a Kelvin–Voigt constitutive model, but leaves $T_{RR}$ free from derivatives of fractional order.

A constitutive equation of fractional order α∈(n−1,n) has the form
(67)$$T_{RR}\left(t\right)=\int_{0}^{\infty }\tilde{M}\left(u\right)\partial_{t}^{n}E^{t}\left(u\right)du.$$

By generality and analogy with the linear viscoelasticity we investigate the thermodynamic consistency of the constitutive equation
(68)$$T_{RR}\left(t\right)=M_{0}E+\int_{0}^{\infty }\tilde{M}\left(u\right)\partial_{t}^{n}E^{t}\left(u\right)du,\;M_{0},\tilde{M}\left(u\right)\in \mathrm{Sym},$$
where $\tilde{M}$ is the kernel possibly of the fractional-order form. By (Equation 66), it follows
$$\psi \left(E,\partial_{t}^{n}E^{t}\right)=\frac{1}{2}E\cdot M_{0}E+E\cdot \int_{0}^{\infty }\tilde{M}\left(u\right)\partial_{t}^{n}E^{t}\left(u\right)du+\hat{\psi }\left(\partial_{t}^{n}E^{t}\right).$$

Hence, we look for the free energy in the form
$$\psi =\frac{1}{2}E\cdot M_{0}E+E\cdot \int_{0}^{\infty }\tilde{M}\left(u\right)\partial_{t}^{n}E^{t}\left(u\right)du-\frac{1}{2}\mu_{n}\int_{0}^{\infty }\partial_{t}^{n}E^{t}\left(u\right)\cdot \tilde{M}\left(u\right)\partial_{t}^{n}E^{t}\left(u\right)du,$$

$\mu_{n}$ being a constant introduced for dimensional reasons. The functional ψ is required to satisfy the inequality (Equation 66)${}_{2}$,
$$d\psi (\partial_{t}^{n}E^{t}|\partial_{t}^{n+1}E^{t})\leq 0.$$

Observe that
$$\begin{array}{c} d\psi =E\left(t\right)\cdot \int_{0}^{\infty }\tilde{M}\left(u\right)\partial_{t}^{n+1}E^{t}\left(u\right)du-\mu_{n}\int_{0}^{\infty }\partial_{t}^{n}E^{t}\left(u\right)\cdot \tilde{M}\left(u\right)\partial_{t}^{n+1}E^{t}\left(u\right)du\\ =\int_{0}^{\infty }\left[E\left(t\right)-\mu_{n}\partial_{t}^{n}E^{t}\left(u\right)\right]\cdot \tilde{M}\left(u\right)\partial_{t}^{n+1}E^{t}\left(u\right)du \end{array}$$
and that
$$\partial_{t}^{n+1}E^{t}\left(u\right)=\partial_{t}^{n}\partial_{t}E^{t}\left(u\right)=-\partial_{u}\partial_{t}^{n}E^{t}\left(u\right)=\frac{1}{\mu_{n}}\partial_{u}\left[E\left(t\right)-\mu_{n}\partial_{t}^{n}E^{t}\left(u\right)\right].$$

Consequently, dψ takes the form
$$d\psi =\frac{1}{\mu_{n}}\int_{0}^{\infty }Q\left(t,u\right)\cdot \tilde{M}\left(u\right)\partial_{u}Q\left(t,u\right)\;du,$$
where
$$Q\left(t,u\right)=E\left(t\right)-\mu_{n}\partial_{t}^{n}E^{t}\left(u\right).$$

We have
$$Q\left(t,u\right)\cdot \tilde{M}\left(u\right)\partial_{u}Q\left(t,u\right)=\partial_{u}\left\{\frac{1}{2}Q\left(t,u\right)\cdot \tilde{M}\left(u\right)Q\left(t,u\right)\right\}-\frac{1}{2}Q\left(t,u\right)\cdot {\tilde{M}}^{\prime }\left(u\right)Q\left(t,u\right).$$

Hence, an integration by parts and the assumption $\tilde{M}\left(\infty \right)=0$ yield
$$d\psi =-\frac{1}{2\mu_{n}}Q\left(t,0\right)\cdot \tilde{M}\left(0\right)Q\left(t,0\right)-\frac{1}{2\mu_{n}}\int_{0}^{\infty }Q\left(t,u\right)\cdot {\tilde{M}}^{\prime }\left(u\right)Q\left(t,u\right)du.$$

Since $\tilde{M}\left(u\right)$ is unbounded as u→0, we cannot write the limit value $\tilde{M}\left(0\right)$. In case of $\tilde{M}$ bounded, the negative definiteness of dψ would imply
(69)$$\frac{1}{\mu_{n}}\tilde{M}\left(0\right)\geq 0,\;\frac{1}{\mu_{n}}{\tilde{M}}^{\prime }\left(u\right)\geq 0.$$

Two aspects are crucial. First, $\tilde{M}\left(0\right)$ bounded is inherently in contrast with the kernels related to the derivatives of fractional order. Furthermore, if $\tilde{M}\left(0\right)$ is bounded, then $\tilde{M}\left(0\right),{\tilde{M}}^{\prime }\left(u\right)$ are both positive or negative definite. This contradicts the view that the influence on the stress of previous strains (or strain–rates) is weaker for those strains that occurred long ago.

## 8. Conclusions

This paper investigates the thermodynamic consistency of three models of linear viscoelasticity. The classical model due to Boltzmann is consistent, only if the Boltzmann function $G^{\prime }$ has a negative half-range sine transform, $G^{\prime }\leq 0$. Moreover, consistent free-energy functionals are subject to the inequality $d\psi (E^{t}|{\dot{E}}^{t})\leq 0$. This in turn is consistent with the proof that a function $-G^{\prime }$ that is decreasing, as is shown by $-G^{\prime \prime }={\left(-G^{\prime }\right)}^{\prime }\leq 0$, has a positive sine transform, $-G_{s}^{\prime }\geq 0$.

The model involving $G^{\prime }\left(u\right)$ in a power law form, $u^{-\beta },\beta \in \left(0,1\right)$, satisfies the required condition $G_{s}^{\prime }\leq 0$. If, instead, the stress depends on the strain–rate history ${\dot{E}}^{t}$, rather than on $E^{t}$, then the required consistency condition should be $G_{c}\geq 0$, and this is satisfied. However, $G^{\prime }\left(u\right)≃u^{-\beta }$ gives an unbounded response to constant histories.

The idea of the dependence on ${\dot{E}}^{t}$ traces back to Pipkin [7] and is of interest in connection with the viscoelastic model of fractional order. For definiteness, a viscoelastic-like constitutive equation is considered in the form (Equation 68). Both the unboundedness of the kernel and the conditions (Equation 69) show that a free-energy functional has still to be determined. For models with derivatives of fractional order, as well as with constitutive equations involving strain–rate histories, finding a free-energy functional consistent with the second law of thermodynamics seems to be an interesting open problem.

## Data Availability

The study did not report any data.

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
