# Peer review of "Thermodynamic Restrictions in Linear Viscoelasticity"

_materials, 2022, doi:10.3390/ma15082706_

Round 1

Reviewer 1 Report

Manuscript materials-1629082 investigates the thermodynamic consistency of linear viscoelastic models. It elaborates three characteristic examples in detail: Boltzmann functions, viscoelastic models with strain rate history and viscoelastic models of fractional order. The manuscript is clearly written and well structured. General statements are confirmed by concrete examples.

Remark to the author:

The paper only considers the thermodynamic consistency of assumptions for Helmholtz free energy. However, a strict mathematical analysis of hyperelasticity also requires further conditions to be fulfilled such as convexity and coercivity. The author should include these aspects and corresponding literature in the introduction or outlook.

Technical remarks

Some cross references for equations on pages 2 and 8 are inactive.

There is a typo in the equation between Eqs. 6 and 7.

Reviewer 2 Report

Summary of the Work

The aim of this work is to study the thermodynamic consistency related to the linear viscoelastic models. More specifically, the author been investigated the Boltzmann classical linear theory, viscoelastic models with strain-rate histories and viscoelastic models of fractional order.

Main obtained results

- For the investigated models, the author established a series of consistency conditions (a synthetic summary of them can be found in Section 8. "Conclusion" (at page 15).

- The author re-proposes the Pipkin idea (1964) in connection with the viscoelastic model of fractional order.

General Comments

- The work is interesting and I enjoyed to read it.

- The author is quite known in the field of rational mechanics.

- No mathematical mistakes have been found. However, some important aspects need to be clarified.

- Some expressions pop-up in the text without a physical explanation. These need to be clarified (please, refer to the suggestions below).

- More recent approaches in non-equilibrium thermodynamics are not quoted in this work (refer to the Sections “Suggestions” below).

- The present work should be better framed by citing, and taking into account, the results that have appeared more recently in the literature. Apart from reference 5., "Fractional Calculus in Bioengineering", the most recent reference cited by the author dates back to about 30 years ago.

Minor

- Several typos have been detected (e.g., the question mark just before Eq. (2) – page 2 etc.). Please check the manuscript.

- Not all the equations have been numbered. This involved some difficulties, leading the reviewer to the use of sentences like "the equation given after Eq.(…) ..." for identifying the expression being referenced.

Suggestions

1) Please, define the meaning of the “bar” in the last equation on page 2.

2) The entropy balance equation is given by Eq.(4). For clarity towards the reader, please explain why the entropy production takes the form ρRr/θ by specifying this contribution in the form flux - force relation.

3) As known, the energy balance equation provided by the expression after Eq.(4) is derived by using the relations-type (i.e., not equal to, but like to)  Gibbs-Duhem equation However, as known these relations rest upon the validity of the "Local Equilibrium Principle" (LEP). The author is asked to clarify this aspect.

4) Section 4. deals with restrictions induced by periodic histories. The author is asked to conciliate his approach with the validity of the LEP.

5) This question is a continuation of the previous one. The author investigated the thermodynamic consistency of the constitutive equations (26), (30) and (37). He provided the restrictions induced by periodic histories. However, it is known that in these conditions, the validity of the flux-force relations, written in local form, is compromised (ref., for instance, to the book of R. Balescu, "Aspects of Anomalous Transport in Plasmas"). Please, discuss this aspect in relation to the validity of the balance entropy equation (4).

We have now reached two crucial issues, subject of the following questions 6) and 7)

6) The history-dependent constitutive equations, written in the form (26), (30) and (37), leads to think that the most appropriate approach to deal with these cases is the well-known Extended Irreversible Thermodynamics (EIT). Indeed, the EIT is a branch of non-equilibrium thermodynamics that goes beyond the local equilibrium hypothesis of classical irreversible thermodynamics. The space of state variables is enlarged by including the fluxes of mass, momentum and energy and eventually higher order fluxes. The consequences induced by this approach are significant (and certainly not negligible) in particular concerning the consistency with the second law of thermodynamics. We may object that the present analysis did not take into account the results established by the EIT. The author is asked to dispel this possible objection.

7) Let's suppose for the moment that somehow it is possible to justify the validity of the local equilibrium principle and, therefore, it is not necessary to apply the EIT formalism. In this work the author analyses the thermodynamic consistency related to the linear viscoelastic models. However, there are recent works that deal with non-linear viscoelastic models (e.g., constitutive equations for magnetically confined plasmas involving non-linear moments, thermodynamical field theories, etc.). If non-linear contributions were taken into account, would the required consistency conditions found by the authors be the same? It is important to answer this question with clarity as we may object that the restrictions identified in this work are not of a physical nature but rather related to the limits of validity of linear viscoelasticity models.

Remark

To the best of my knowledge, models with derivatives of fractional order as well as with constitutive equations involving strain-rate histories are currently proposed for treating magnetically confined plasmas in turbulent regime (ref., for example, to the above cited book). However, it is well-established that in these condition the LEP is violated. Hence, finding a free energy functional consistent with the second law of thermodynamics is certainly an interesting problem, but it is suggested to consider the above.

Conclusions

On first reading the work looks flawless. But then, looking more deeply into the work, particularly the physics behind each basic equation, several issues emerge that need to be clarified. I agree that the work is of quality and deserves to be published but, written in this form, it leaves too many doubts to the reader, especially to the scientists working on thermodynamics of irreversible processes. The author is therefore invited to take into account the suggestions expressed above. In my opinion, this will contribute to attract more readers.

Reviewer 3 Report

In this manuscript, the author aims at investigate the relation between Boltzman like viscoelastic models but formulated at finite strain and the standard thermodynamic framework of continuum mechanics. The objective of the author is to determine which thermodynamics restriction have to applied to theses models in order to be consistent with thermodynamics.

To the reviewer point of view, the only originality of this manuscript is to propose a discussion for strain-rate history dependent models. The other cases are already discussed in the literature and the main results are already known. This manuscript suffers from several drawbacks and needs to be improved before being considered for publication. The reviewer suggests to have a look at the following points:

  • Major remarks:

    • The author must improve the literature overview. Similar results than the one obtain by the author (G’>0 and G’’<0) have been obtained by other authors years ago. The for instance the works of Alexander Lion and Peter Haupt: On finite linear viscoelasticity of incompressible isotropic materials, Acta Mechanica 159, 87-124, 2002

    • In section 2 there is a mixture between Eulerian and Lagragian formalisms. This is unnecessary and leads to errors and confusions. The infinitesimal strain measure in the Eulerian configuration is not infinitesimal at all and not consistent with thermodynamics contrarily to the Green-Lagrange strain which is used everywhere in the following. The author should remove Eulerian definitions and directly use a Lagrangian formulation (for instance eq. (1) can not leads to eq. (2), this is two different models).

    • The author assume that the relaxation kernel is bounded for a given norm at eq. (3). To the reviewer point of view this hypothesis is unnecessary.

    • At the end of page 2 the arguments of the author to linearize the strain makes no sense. At small strain we have to assume that F=1+h where h<<1 we therefore neglect every second and higher order terms for h. This linearization is not used in the manuscript so why mention it?

    • In eq. (2) why the author assume a linear relation between the Piola equilibrium stress (the one related to G0) and the strain? This is not a necessary hypothesis from the reviewer point of view and this equilibrium part can be more generally defined from a hyperelastic model.

    • In section 3, what is t0? The author mentions the time of constant histories: what does this correspond to? Why is this needed?

    • The author states that the Coleman-Noll inequality (eq. 5) implies an inequality on the strain energy. To the reviewer point of view this is a non sense. Eq. 5 only states that the dissipation is always positive or null but does not impose restriction on level of energy for different strain histories. Coleman-Noll inequality allows multiple equilibrium solutions and instabilities (non-convex free energies).

    • To the reviewer point of view, a major difficulty in the understanding of the author’s approach is due to the definition of a strain history, Et(s), that is assumed to have an evolution that is independent of the evolution of current strain E(t). The approach is similar to local (in time) models with internal variables but here the author does not used the thermodynamics to obtain an evolution equation for the supplementary internal variable. It is only used to derive thermodynamics constraint. However many details are obscure. First it is not clear that D_t(Et(s)) and D_t(E(t)) are independent, second the calculus of the Frechet derivative at eq. (9) seams wrong, third how is obtain inequality at eq. (10)? fourth: what is psi_G? To what it correspond? etc.

    • The derivation of condition at eq. (11) is unclear: what is the physical meaning of having a free energy function that is minimum in the case Et(s)=E(t) ? In this case it can be seen from the definition of psi_G or psi that only the elastic part remain in the free energy, therefore how is obtain eq. (11)?

    • The integration by part before eq. (12) is not clear: why only the G’(s) is derived in the second term (and no the strain)?

    • The author has chosen to consider non isothermal situations but do not exploit physical restriction that should applied on the relaxation kernel if one consider the energy balance for transient thermal situations. Why not consider only the isothermal and adiabatic case in this manuscript ?

    • Results obtained at eq. (17) and (19) are obvious: G0 and Ginfty are elasticity tensors in the Lagrangian formulation and therefore these tensors have major and minor symmetry whatever is the loading (not only in the case of periodic histories).

  • Minor corrections:

    • All equations should be numbered

    • Reference to equation in the text are not always consistent

    • Page 2 second paragraph: “D=SymL is the stretching tensor” should be replaced by D=SymL is the eulerian rate of deformation

    • The sense of inequalities at eq. (11) and eq. (12) should be clarify: to the reviewer point of view this tensorial inequalities makes no sense (I guess that the author think about positive or negative definitiveness of the tensors).

Round 2

Reviewer 2 Report

The author has satisfactorily answered the questions raised in my previous report. My main concerns were two:
1) Update references;
2) Taking into account the latest results obtained in this research area and, in particular, the new thermodynamic theories in the field of irreversible processes such as the Extended Irreversible Thermodynamics (EIT) (applicable in case of the violation of the local equilibrium hypothesis) and the of Thermodynamical Field Theory (TFT) (applicable in case of nonlinear closure relations).
The author has made an effort to fill in the above gaps, which I really appreciate. I also noticed that some unnecessary assumptions of the author, which appeared in the previous version of the work, have been eliminated. In my opinion, this new version of the manuscript deserves to be published.